# A Comparison of Different Approaches to Clinical Phenotyping of Lithium Response: A Proof of Principle Study Employing Genetic Variants of Three Candidate Circadian Genes

**DOI:** 10.3390/ph14111072

**Published:** 2021-10-23

**Authors:** Jan Scott, Mohamed Lajnef, Romain Icick, Frank Bellivier, Cynthia Marie-Claire, Bruno Etain

**Affiliations:** 1Institute of Neuroscience, Newcastle University, Newcastle NE7 6RU, UK; jan.scott@newcastle.ac.uk; 2Sorbonne Paris Cité, Université Paris Diderot, F-75006 Paris, France; 3INSERM UMR 955, IMRB, Université Paris Est Créteil, F-94000 Créteil, France; mohamed.lajnef@inserm.fr; 4INSERM UMR-S 1144, Université de Paris, F-75006 Paris, France; romain.icick@lrb.aphp.fr (R.I.); frank.bellivier@inserm.fr (F.B.); cynthia.marie-claire@parisdescartes.fr (C.M.-C.); 5AP-HP Nord, DMU Neurosciences, GHU Lariboisière-Fernand Widal, Département de Psychiatrie et de Médecine Addictologique, F-75010 Paris, France

**Keywords:** bipolar disorder, lithium, response, phenotype, genetics, circadian genes, machine learning

## Abstract

Optimal classification of the response to lithium (Li) is crucial in genetic and biomarker research. This proof of concept study aims at exploring whether different approaches to phenotyping the response to Li may influence the likelihood of detecting associations between the response and genetic markers. We operationalized Li response phenotypes using the Retrospective Assessment of Response to Lithium Scale (i.e., the Alda scale) in a sample of 164 cases with bipolar disorder (BD). Three phenotypes were defined using the established approaches, whilst two phenotypes were generated by machine learning algorithms. We examined whether these five different Li response phenotypes showed different levels of statistically significant associations with polymorphisms of three candidate circadian genes (*RORA*, *TIMELESS* and *PPARGC1A*), which were selected for this study because they were plausibly linked with the response to Li. The three original and two revised Alda ratings showed low levels of discordance (misclassification rates: 8–12%). However, the significance of associations with circadian genes differed when examining previously recommended categorical and continuous phenotypes versus machine-learning derived phenotypes. Findings using machine learning approaches identified more putative signals of the Li response. Established approaches to Li response phenotyping are easy to use but may lead to a significant loss of data (excluding partial responders) due to recent attempts to improve the reliability of the original rating system. While machine learning approaches require additional modeling to generate Li response phenotypes, they may offer a more nuanced approach, which, in turn, would enhance the probability of identifying significant signals in genetic studies.

## 1. Introduction

Clinical practice guidelines identify lithium (Li) as a first-line treatment for mood stabilization in bipolar disorders (BD) [1,2]. Unfortunately, only approximately 30% of patients show a good response, and variability in treatment outcome is poorly understood [3,4,5]. It is envisioned that precision medicine or personalized psychiatry approaches will enable the stratification of BD cases into treatment-relevant subgroups [6,7]. However, for this research to be successful, greater attention is needed regarding the method for classifying clinical phenotypes of the Li response [8].

The ideal research assessment of the Li response would involve the systematic prospective follow-up of Li-naive cases that are prescribed this medication for the first time [9]. However, this gold-standard approach is complex, so most genetic studies [10,11,12] identify clinical phenotypes of the Li response from ratings of the Retrospective Assessment of Response to Lithium Scale (usually referred to as the Alda scale) [13]. The Alda scale comprises two subscales: The A scale (which measures overall response) and the B scale (which assesses five potential confounders of response). In the original guidelines, Li response was reported either by the Total Score as a continuous measure (TS = A score minus B score) or, more often, as a categorical outcome (with cases classified as good or non-responders, i.e., GR or NR) [13,14]. However, when Manchia et al. (2013) undertook an inter-rater reliability study with researchers from the Consortium on Li Genetics (ConLiGen), reliability was low for Alda scale ratings of BD cases with high B scale scores (typically cases with complex clinical presentations). It was suggested that in order to overcome these problems, the Li response (using the A scale) should only be rated in the subsample of individuals with a low score on the B scale [15].

More recently, we examined alternative approaches to improving the performance of the Alda scale [16]. We systematically assessed its clinimetric and psychometric properties (in a ConLiGen sample N > 2500) and demonstrated that the Alda scale is best viewed as a multi-dimensional index that assesses several independent modifiers of the noise-to-signal ratio for Li response (i.e., rating the Alda scale may not be a simple procedure). Applying this knowledge, we implemented a stepwise algorithm (generated via machine learning) to produce a best estimate of the Li response. When the algorithm was used in separate, independent clinical samples, it was shown that this approach can increase the overall accuracy of the allocation of BD cases to categorical or continuous Li response phenotypes without sample attrition [16,17]. Furthermore, this best estimate classification identified clinical characteristics and objective markers of rest–activity rhythms associated with Li response [16,18]. The next step in assessing the validity of this new strategy for ascertaining Li response phenotypes is to undertake a “proof of principle” study that compares established and revised Alda scale ratings and examines their associations with genetic markers for Li response.

To select plausible genes for this proof of principle study, we decided to focus on genes involved in the circadian system. We argue that this is justified as it is repeatedly suggested that the therapeutic action of Li can be partly explained by modifications and stabilization of circadian rhythms [19,20]. For instance, recent meta-analyses and reviews suggest that Li modifies the phase preference and amplitude of rhythms in BD cases and that genetic polymorphisms associated with circadian rhythm dysregulation contribute to the Li response [21,22,23,24,25]. More specifically, our own research group identified significant associations between certain *TIMELESS* and *RORA* genotypes and circadian phenotypes (phase preference and amplitude of rhythms) in BD cases and found that the Li response may be associated with some *PPARGC1A* and/or *RORA* genotypes [26,27]. Other publications similarly identify that levels of expression and/or polymorphisms of core clock genes (such as *TIMELESS*) may be associated with the level of Li response [28,29]. Given these findings, we hypothesized that there may be significant associations between the Li response and these three circadian/clock genes (*RORA, TIMELESS* and *PPARGC1A*) and that these genes may be plausible candidates for this proof of principle study.

In sum, although the Alda scale is the most widely employed rating of Li response, there is a lack of consensus on how best to ascertain clinical phenotypes, which potentially hinders progress in this research field. This paper describes a database study of Li response that explores the strengths and weaknesses of different approaches to rating the Alda scale in a well-characterized clinical sample of adults with a diagnosis of BD. We report overall agreement between different approaches to evaluating Li response phenotypes and rates of discordance. Then, we examine associations between each clinical phenotype for Li response and genetic variants in three candidate circadian genes. Finally, we consider the advantages and disadvantages of the different approaches.

## 2. Results

### 2.1. Sample Characteristics and Li Response Phenotypes

As shown in Table 1, most of the sample was female (60%) and met diagnostic criteria for BD-I (78%). Mean age at interview was approximately 45 and mean duration of illness was just under 20 years. Regarding the Alda scale, the mean A scale score was 6.33; for the B scale, the modal score was zero for items B1-3 and one for items B4-5.

According to the original approaches, 21% (*n* = 35) of cases were classified as GR according to the original categorical approach (Alda Cats) and, using a continuous measure of Li response, the mean TS (Total Score) was 3.7 (s.d 2.8; median = 4). Using the approach proposed by Manchia et al. 2013 for estimating Li response, we found that there were 106 cases with a B score < 4; in this subgroup, the mean A scale score was 6.9 (s.d. 2.8; median = 8). Machine learning classified 26% cases (*n* = 43) as GR (using the categorical “Algo” approach); the nearest equivalent of the continuous measurement of Li response, namely the probability of GR (GRp), was estimated as GRp > 0.62.

### 2.2. Comparison of Accuracy and Discordance for Li Response Phenotypes

When we compared traditional and machine learning approaches to classification (i.e., Alda Cats versus Algo, TS versus GRp and A/Low B versus GRp), we found that the PPVs were all >80%, the NPVs were all >95% and overall agreements were all >90%. The proportion of cases with discordant classifications was lowest for categorical phenotypes (8%) and highest for A/Low B versus GRp (12%). The latter was likely influenced by the reduced sample size (as only 106 cases met the A/Low B criteria).

### 2.3. Associations between Genotypes and Li Response Phenotypes

As shown in Figure 1, the A/Low B phenotype showed no associations with any of the studied SNPs (there was a trend with *TIMELESS*); TS showed a significant association with *TIMELESS*, whilst Alda Cats showed significant associations with *TIMELESS* and *PPARGCIA*. The Algo classification showed significant associations with all three SNPs, whilst GRp showed associations with *TIMELESS* and *PPARGC1A* (with a trend for *RORA*).

The classification trees for Li response categories are shown in Figure 2. As can be seen in Figure 2a, only the *TIMELESS* genotype met the criteria for inclusion in the tree based on Alda Cats (X^2^ = 21.1; Adjusted *p* value < 0.001). Overall, 89% of *TIMELESS* GG homozygotes were NR (Node 1), whilst 43% of those with *TIMELESS* AA/AG genotypes were GR (Node 2). The most obvious difference in the classification tree (Figure 2b) based on the Algo is that both the *RORA* (X^2^ = 12.46; Adjusted *p* value < 0.002) and *TIMELESS* (X^2^ = 5.03; Adjusted *p* value < 0.02) genotypes were included in the model (i.e., including *PPARGC1A* did not further improve the classification). For *RORA*, 91% of CC homozygotes were NR (Node 1), while 47% of TT homozygotes were GR (Node 3). Among individuals with a *RORA* TC genotype (Node 2), 80% of *TIMELESS* GG homozygotes were NR (Node 4), whilst 43% of individuals with a *TIMELESS* AA/AG genotype were GR (Node 5).

For readers less familiar with CHAID (Chi-Square Automatic Interaction Detector) analysis, we briefly summarize here how to interpret the outputs. A CHAID analysis generates a bifurcating decision tree composed of a root node (the variable with the strongest association with the dependent variable and lowest *p* value), which then branches and grows iteratively into internal and terminal nodes (the latter represent variables that carry the maximum information). The order of importance of explanatory variables is represented by the tree structure and the percentages shown within the nodes are an indicator of the relevance of each characteristic as a primary predictor of Li response status (similar to PPV and NPV). In the current study, the classification rates are reported for different combinations of genotypes. As the roots and nodes may link in different ways (following an “if–then” type sequence), there may be more than one classification rate associated with variables included in the tree. Tree building ends when the *p* values of all the observed independent variables are above the specified threshold for statistical significance, so the absence of any genotype or demographic variables from the trees that we report indicates that those items (although included in the list of variables for analysis) did not make a meaningful additional contribution to case classification. We have not reported summary statistics for each root and node (this has been done to make the classification tree easier to interpret, but all these statistics are available upon request).

## 3. Discussion

The primary goal of this study was not to interpret any associations between genetic variants of circadian genes and Li response per se (nor to discuss any biological relevance of these genes), but to examine if machine learning approaches to clinical phenotyping are viable and have any advantages over established strategies. This is important as it is increasingly acknowledged that advances in precision psychiatry need an integrated science approach to ensure reliable and valid ascertainment of clinical phenotypes and of any differential associations with genetic and/or other type of biomarkers of disease or treatment outcome [4]. In this proof of principle study, we explored links between clinical phenotypes of Li response (three original and two revised estimates) and genetic variants in three candidate circadian genes. The latter were selected for their involvement in the molecular mechanisms of the regulation of circadian rhythms and their previous associations with BD and/or treatment response and thus were plausible candidates to be used in a proof of principle study.

We demonstrated that, although discordance rates for case classification between the different approaches to phenotyping were low, the subtle shifts in the balance between GR and NR may enable the revised approaches to identify more potential genetic signals of Li response than the traditional approaches. Further, an exploratory data mining analysis (using CHAID analysis) identified a subtle inter-relationship between genotypes (especially those of *TIMELESS* and *RORA*) and the revised categorical (Algo) phenotype that was not discovered by the original (Alda Cats) approach to the classification of responders.

Of course, our findings must be treated with caution as the study population was recruited from a limited number of academic psychiatry clinics in the same country and represents a convenience sample extracted from a pre-existing dataset. Importantly, the selection and number of candidate genes and SNPs may be questioned, as we focused on one candidate biological pathway and used only one SNP per gene, which is obviously restrictive and likely biased. This selection of candidate genes is, by definition, debatable and other approaches may have focused rather on genes that reached (or nearly reached) significant thresholds in previous GWAS studies [30,31]. Moreover, this sample may be regarded as too small to detect any differences between genotype-based groups for response to Li. However, a design with group sample sizes (based on genotype distribution) can detect effect sizes of 0.5 for *TIMELESS* and *PPARGC1A* and an effect size of 0.6 for *RORA*, with a power above 0.80, assuming a two-sided criterion for detection (based on means and SDs of the Alda total score) that allows a maximum Type I error rate of alpha = 0.05. Despite these limitations, the potential strength of this study is that it highlights that the three established and most widely used approaches to operationalizing the Li response do not produce consistent signals. This is important as nearly all genetic studies of the Li response have reported their findings based on the Alda Cats approach alongside one of the two continuous measures [10]. The disparities in findings across these three traditional response phenotypes are a cause for concern and, whilst imperfect, the revised algorithms do show greater consistency.

Of the three original approaches, the A/Low B strategy is the newest estimate of Li response, and it was introduced because of concerns over the accuracy of the TS and, by default, of the Alda Cats [15]. It can be argued that the A/Low B approach is justifiable as (a) it is easy to implement and was introduced to enhance inter-rater reliability, and (b) it is likely to minimize false positives. However, excluding cases with high B scale scores can adversely impact treatment research as (a) it reduces the sample size for investigation (e.g., 34% of the current sample were excluded from analyses using this approach and there was a clear drop of -log(p) as compared to TS), and (b) it assumes that all confounders are equally important across all samples (which other research indicates is unlikely). As such, this estimate represents a pragmatic rather than empirical approach to trying to overcome some of the psychometric weaknesses of the Alda scale. In the current study, this approach produced results that are difficult to reconcile with findings associated with other established approaches (Alda Cats and/or TS) and failed to identify signals identified by the machine learning approaches.

The most obvious advantage of the best estimate approach to phenotyping is that it offers a more nuanced approach to defining the Li response as the machine learning algorithms address the differential impact on response (or confidence in assessing response) of some confounders and/or the complexity of inter-relationships between confounders within a given study population. The Algo classification is easier to replicate and interpret, as it balances GR versus NR. Further, the Algo and GRp approaches appear to show more similarities than differences (in contrast to original approaches). However, we believe that the model for generating GRp requires more work (i.e., it probably needs further refinement of thresholds and/or greater consideration of other confounders and/or their inter-relationships, with a broader range of demographic and clinical factors than those currently considered by the Alda scale). Overall, the main advantage of the best estimate approach is that, unlike the ‘A/Low B’ strategy, the GR/NR split is empirically derived, and the algorithm attempts to classify all cases without exception (also, thresholds for GRp could be modified according to study priorities, e.g., preference for identifying true GR or true NR).

At a practical level, the machine learning approaches to evaluating the Li response can be applied in two ways. For investigators with limited resources, existing machine learning algorithms can be applied to generate Li response phenotypes (by running existing statistical syntax derived from ConLiGen samples; [16,30]). Alternatively, researchers with more time and resources could run the machine learning model de novo and generate Li response phenotypes that are specific to their BD study population. This can provide insights into how sampling influences the identification of Li response phenotypes and may help in the discovery of linked biomarkers in datasets with genomic data [32], brain imaging [33] or other types of biomarkers [34].

In conclusion, we note that the original TS/Alda Cats approaches to rating the Alda scale are somewhat simplistic. For instance, it fails to address the issue of Li non-response due to minimal direct benefit from Li (A score rating) versus non-response associated with high levels of confounding (e.g., those with high B and high A score versus those with high B and low A score, etc.). The A/Low B approach has some advantages, not least that it can be efficiently applied. However, this more stringent approach leads to a reduction in sample size. This might be accommodated in large studies, but it is a significant issue in smaller-scale studies. Furthermore, this approach actively deselects cases with high B scores (which, as we know, often have complex presentations). This may be appropriate for signal detection in genetic research, but it undermines clinical research aimed at understanding the Li response in difficult-to-treat cases (i.e., those that often require the most input and resources). The latter represent a real-world clinical population where response prediction would be highly valued. The next step for the current project is to replicate the findings in a larger study designed with the specific aim of testing the revised approaches to phenotyping in a representative clinical cohort, at the level of the entire circadian system genes and/or at a genome-wide level.

## 4. Materials and Methods

The study received ethical approval from the French Ethics and Data Protection and Freedom of Information Commissions (CPPRB, RCB:2008-AO14-65-50). Here, we briefly outline the methodology; full details regarding machine learning, genotyping procedures and analyses are available elsewhere and/or are summarized in the published protocol [16,17] (ClinicalTrials.org: NCT02627404).

### 4.1. Sample

The study uses de-identified data from 164 adults aged >18 years who gave written informed consent to participate in a study of Li response and provided a blood sample for genotyping. Study participants were unrelated individuals of Caucasian origin, who had a diagnosis of BD that met DSM-IV criteria [35] according to the French version of the Diagnostic Interview for Genetic Studies [36,37] and who were in remission at the time of recruitment (>=3 months since the last major mood episode) [38] and currently euthymic according to the MADRS (Montgomery Asberg Depression Rating Scale) and the YMRS (Young Mania Rating Scale) [39,40].

### 4.2. Phenotyping

Lithium response was estimated from ratings of the two subscales (A and B) of the Alda scale [13]. The A scale assesses change in illness activity whilst receiving Li (which represents the clinically assessed change in frequency, severity and duration of episodes), with response rated on a 0–10 continuum and a higher A scale score indicative of better response. The B scale items are all rated 0–2. Each item measures a clinical characteristic that may attenuate or confound response, namely B1—number of episodes prior to Li (a score of 2 suggests fewer episodes, making judgements about the impact of Li more difficult); B2—frequency of episodes prior to Li (a score of 2 suggests low frequency); B3—assesses duration of Li treatment (a score of 2 would suggest a short period of time, making judgements about benefits more difficult); B4—measures adherence with Li (a score of 2 indicating poor adherence) and B5—assesses complexity of treatment regime, including polypharmacy and co-prescription of mood stabilizers (a high score indicating a more complex treatment regime).The Li response phenotypes are operationalized as follows:
(a)Original approaches to rating the Alda scaleThe three most widely used approaches are:-Total Score (TS): a continuous measure represented by the TS (A scale minus B scale score); if B > A, then the TS is reported as zero.-Original classification (Alda Cats): Li response categorized as GR (TS >= 7) or NR (TS < 7).-A score in cases with a low B scale score (A/Low B): Li response is represented by the A scale score (continuous variable), but assessment is restricted to individuals with B < 4; those with high B scores are excluded from the analysis.(b)Machine learning approach to rating the Alda scaleIn the best estimate classification approach, a machine learning algorithm determines a set of “if–then” rules for determining the probability of GR and NR. The analysis enters the B scale item scores in a sequence; this usually starts with treatment complexity (adherence and polypharmacy), then duration of Li treatment and/or illness activity (the exact sequence and combination of item scores is generated by the machine learning model). The algorithm stops running once the optimal classification is reached, irrespective of whether all B items have been included (for details, see [16]). Here, we report the findings on Li response phenotypes as a categorical measure (New Algorithm; Algo). To create a continuous measure to compare with TS and A/Low B, we also estimated GRp (a measure of probability of GR in this sample).

### 4.3. Genotyping, Quality Control and Selection of Polymorphisms in RORA, PPARGC1A and TIMELESS

Genomic DNA samples were extracted from peripheral blood leukocytes or B-lymphoblastoid cell lines by standard procedures. Genotyping was performed at the Centre National de Genotypage (CNG, Paris, France) using HumanHap550 or 610-Quad Beadchips (Illumina Inc., San Diego, CA, USA.). Analyses were performed using PLINK v1.07 30 [41]. All available single-nucleotide polymorphisms (SNPs) within *RORA*, *PPARGC1A* and *TIMELESS* and within 10 kilobase pairs (Kbp) upstream and downstream from the coding sequence (extracted from the RefSeq Database (National Center for Biotechnology Information; https://www.ncbi.nlm.nih.gov/refseq/, accessed on 18 October 2021)) were used to explore exonic and intronic regions, as well as cis-regulatory regions. SNPs were included in the following association analysis if they fulfilled the following quality criteria (assessed using PLINK software, v 1.9): (1) minor allele frequencies (MAF) greater than or equal to 5%; (2) genotyping call rate for at least 97% of SNPs; (3) call rate averaged 90% and (4) SNP heterozygosis between (m−3 s.d.) and (m+3 s.d.). In order to select the SNPs, we used PLINK to determine the associations between SNPs and the Alda total score. Briefly, for each gene, the program: (1) determines which SNPs are in linkage disequilibrium (LD) (above a certain threshold, fixed in this study at r2 = 0.05); (2) performs a single SNP association analysis using a non-parametric approach; and (3) determines the associated SNP having the highest statistical significance (selection of the “best SNP” option). This method allowed us to identify, among all available SNPs within a given gene, which SNP was the most strongly associated with the phenotype (whatever the level of significance). Among all available SNPs in the three selected genes (*RORA n* = 140; *PPARGC1A n* = 25; and *TIMELESS n* = 8), this approach retained rs17204910 in *RORA*, rs2932965 in *PPARGC1A* and rs774045 in *TIMELESS*. For these three SNPs, all genotypes were in Hardy–Weinberg equilibrium.

### 4.4. Statistical Analysis

First, we compared estimates of Li response using the original and new approaches to rating the Alda scale, reporting the positive and negative predictive values (PPV, NPV), the overall accuracy and discordance rates. For the purposes of the analyses, we assumed that the original ratings represent the “gold standard” (i.e., for categorical outcomes, false positives are cases that were classified as GR according to the new algorithms but not the original rating). The classification obtained for Alda Categories was compared with Algo, whilst the A score/Low B measure was compared with GR according to the Algo (with analyses undertaken using the program that is publicly available on the Oxford University evidence-based medicine website: https://www.cebm.ox.ac.uk, accessed on 18 October 2021). To interpret the findings, we used the indicators established for diagnostic test comparisons used in clinical settings, which suggested that we could expect the new Alda ratings to show PPV, NPV and accuracy estimates of >80–85% (compared with established ratings).

Associations between genotypes of *TIMELESS* (GG versus GA/AA), *RORA* (CC versus TC versus TT) and *PPARGC1A* (GG versus GA/AA) and Li response phenotypes are reported as −log_10_(p), and levels of statistical significance are reported as *p* < 0.017 (corrected for 3 genes) and *p* < 0.003 (corrected for 3 genes and 5 phenotypes).

Next, for categorical classifications (Alda Cats and Algo), we employed Chi-Square Automatic Interaction Detector (CHAID) analysis to explore whether any combinations of genes improved the ascertainment of GR or NR cases. This analysis generated a classification tree, which represents a sequential model consisting of a set of if–then rules for the partition of heterogenous input data into groups that are homogenous regarding the dependent/outcome variable categories. To avoid overfitting of CHAID, we adjusted the model for age and sex (i.e., known variables of influence that were not considered already within the Alda rating) and analyses were cross-validated. In the figures shown, the order of importance of explanatory variables is explicitly represented by the tree structure, and tree building ended when the *p* values of all the observed independent variables were above the specified threshold for statistical significance (usually, an alpha level of 0.05, corrected for the number of statistical tests within each predictor using a Bonferroni multiplier that adjusted all *p* values for multiple testing).

## 5. Conclusions

Established approaches to Li response phenotyping are easy to use but may lead to a significant loss of data (excluding partial responders) due to recent attempts to improve the reliability of the original rating system. While machine learning approaches require additional modeling to generate Li response phenotypes, they may offer a more nuanced approach, which, in turn, would enhance the probability of identifying significant signals in genetic studies.

## Figures and Tables

**Figure 1 pharmaceuticals-14-01072-f001:**
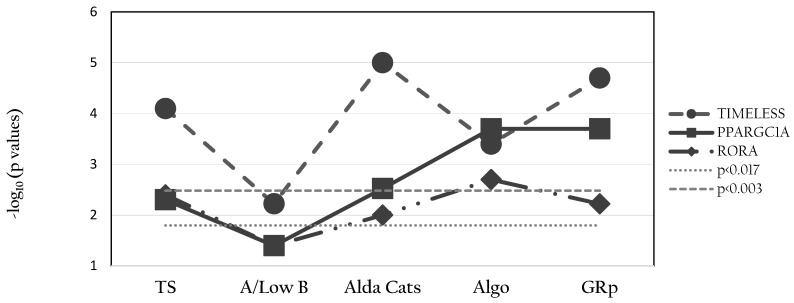
Examination of Li response phenotypes and SNPs within *TIMELESS*, *PPARGC1A* and *RORA*. (An A-dominant model was used for *TIMELESS* and *PPARGC1A*).

**Figure 2 pharmaceuticals-14-01072-f002:**
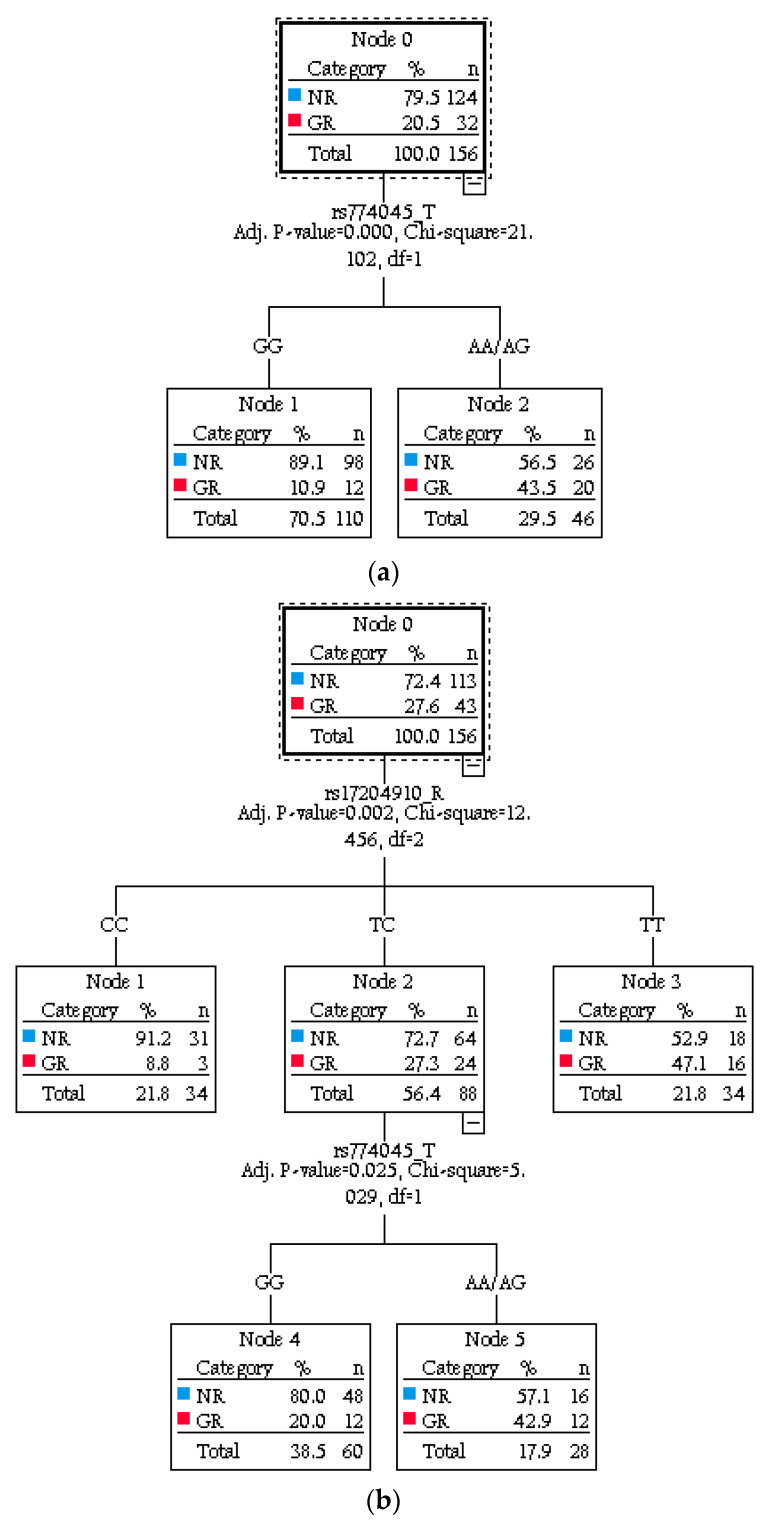
Classification tree models for Li response phenotypes (NR = non-response; GR = good response) and genotypes of candidate circadian genes (rs17204910-R: *RORA*; rs774045-T: *TIMELESS*). (**a**) Classification tree using the original categories (Alda Cats). (**b**) Classification tree using the machine learning algorithm (Algo).

**Table 1 pharmaceuticals-14-01072-t001:** Sample characteristics (*n* = 164, unless otherwise stated).

Characteristic	Mean (s.d.) or Number (% *)
**Demography and diagnosis**	
Female	99 (60%)
Mean age at interview in years	44.70 (12.29)
Bipolar Disorder type I	128 (78%)
Mean duration of illness in years	19.43 (11.27)
**Alda Scale scores**	
A scale score	6.33 (2.99)
**B scale items: prevalence of raw scores ****	
B1—Number of episodes pre-Li	112 (68%): 45 (27%): 8 (5%)
B2—Frequency of episodes pre-Li	100 (61%): 57 (34%): 8 (5%)
B3—Duration Li treatment	122 (74%): 15 (9%): 28 (17%)
B4—Adherence to Li	18 (11%): 140 (85%): 7 (4%)
B5—Co-prescriptions/Polypharmacy	57 (34%): 63 (39%): 45 (27%)
**Genotypes *****	
*RORA* (rs17204910)	CC: 34 - TC: 88 - TT: 34
*PPARGC1A* (rs2932965)	AA: 23 - AG: 79 - GG: 53
*TIMELESS* (rs774045)	AA: 1 - AG: 45 - GG: 110

s.d.: standard deviation; Li: lithium; * reported to the nearest whole number; ** B items are scored as 0:1:2 (high score indicates more confounding); *** *N* = 156.

## Data Availability

Data is contained within the article.

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
