# Peer review of "A Comparison of Different Approaches to Clinical Phenotyping of Lithium Response: A Proof of Principle Study Employing Genetic Variants of Three Candidate Circadian Genes"

_pharmaceuticals, 2021, doi:10.3390/ph14111072_

Round 1

Reviewer 1 Report

This interesting manuscript examines whether different approaches to clinical phenotyping influence the likelihood of detecting associations between Li response and polymorphisms of three genes plausibly linked with treatment response in bipolar disorders. The paper is well-written, and the data are exhaustively presented. The clinical implications of the study are appropriately discussed.

Author Response

Comment 1

This interesting manuscript examines whether different approaches to clinical phenotyping influence the likelihood of detecting associations between Li response and polymorphisms of three genes plausibly linked with treatment response in bipolar disorders. The paper is well-written, and the data are exhaustively presented. The clinical implications of the study are appropriately discussed.

Response 1

We thank the reviewer for this positive review.

Reviewer 2 Report

The authors present a well-written description of a proof of concept/principle study describing the use of machine learning vs established methods to classify lithium response phenotypes. These phenotypes were then used to identify associations between candidate SNPs and response with the various phenotypes.

The authors also present a careful discussion about the potential utility of their approach in precision medicine research, as well as a thoughtful overview of limitations of their work.

I appreciate their breakdown of the Alda scale scoring of the B criterion, but I think it would be helpful to include (as a supplement or within the main text), the actual questions and response options so readers have easy reference to what a 0-2 means and how that reflects differences in adherence (as an example). 

Author Response

The authors present a well-written description of a proof of concept/principle study describing the use of machine learning vs established methods to classify lithium response phenotypes. These phenotypes were then used to identify associations between candidate SNPs and response with the various phenotypes. The authors also present a careful discussion about the potential utility of their approach in precision medicine research, as well as a thoughtful overview of limitations of their work.

 Comment 1

I appreciate their breakdown of the Alda scale scoring of the B criterion, but I think it would be helpful to include (as a supplement or within the main text), the actual questions and response options so readers have easy reference to what a 0-2 means and how that reflects differences in adherence (as an example). 

Response 1

We described this approach in more detail in an earlier draft of the paper, but we had abbreviated this section (to reduce the word count). However, we are happy to re-introduce the extended text (into the methods section), which now states:

The A scale assesses change in illness activity whilst receiving Li (which represents the clinically assessed change in frequency, severity, and duration of episodes), with response rated on a 0-10 continuum and a higher A scale score indicative of better response. The B scale items are all rated 0-2. Each item measures a clinical characteristic that may attenuate or confound response, namely: B1- number of episodes prior to Li (a score of 2 suggests fewer episodes, making judgements about the impact of Li more difficult); B2- frequency of episodes prior to Li (a score of 2 suggests low frequency); B3- assesses duration of Li treatment (a score of 2 would suggest a short period of time, making judgements about benefits more difficult); B4- measures adherence with Li (a score of 2 indicating poor adherence) and B5- assesses complexity of treatment regime, including polypharmacy and co-prescription of mood stabilizers (a high score indicating a more complex treatment regime).

Reviewer 3 Report

A comparison of different approaches to clinically phenotyping of lithium response: a proof of principle study employing genetic variants of three candidate circadian genes

Abstract is long and complicated, needs to be simplified.

Introduction:

Very well written.

With respect to the genes chosen, why did the authors choose to ignore the genes discovered in the ConLiGen- GWAS – (https://www.ncbi.nlm.nih.gov/pmc/articles/PMC5179929/)? I would have assumed these would have been assessed.

Methods;

Lines 337 – 338: Please provide more detail on how the PPV and NPV values were generated.

Lines 332 – 335- Why were only three SNPs retained?

Would be good to include a discussion on sample size and power calculation

Results & Discussion

Well compiled. I appreciated the results expansion on explanation of CHAID. The discussion was well written and not overarching with respect to conclusions.

Author Response

Comment 1

Abstract is long and complicated, needs to be simplified.

Response 1

We have shortened and simplified the abstract as shown below.

Revised Abstract:

Background: Optimal classification of response to lithium (Li) is crucial in genetic and biomarker research. This proof-of-concept study aims at exploring whether different approaches to phenotyping response to Li may influence the likelihood of detecting associations between response and genetic markers.

Methods: We operationalized Li response phenotypes using the Retrospective Assessment of Response to Lithium Scale (i.e., the Alda scale) in a sample of 164 cases with bipolar disorders (BD). Three phenotypes were defined using the established approaches, whilst two phenotypes were generated by machine learning algorithms. We examined whether these five different Li response phenotypes showed different levels of statistically significant associations with polymorphisms of three candidate circadian genes (RORA, TIMELESS and PPARGC1A), that were selected for this study because plausibly linked with response to Li.

Results: The three original and two revised Alda ratings showed low levels of discordance (misclassification rates: 8-12%). However, significances of associations with circadian genes differed when examining previously recommended categorical and continuous phenotypes versus machine-learning derived phenotypes. Findings using machine learning approaches identified more putative signals of Li response.

Conclusions: Established approaches to Li response phenotyping are easy to use but may lead to significant loss of data (excluding partial responders) due to recent attempts to improve the reliability of the original rating system. While machine learning approaches require additional modelling to generate Li response phenotypes, they may offer a more nuanced approach which in turn would enhance the probability to identify significant signals in genetic studies.

 Comment 2

Introduction: Very well written.

Response 2

We thank you for this comment.

 Comment 3

With respect to the genes chosen, why did the authors choose to ignore the genes discovered in the ConLiGen- GWAS (https://www.ncbi.nlm.nih.gov/pmc/articles/PMC5179929/)? I would have assumed these would have been assessed.

Response 3

We wish to highlight that our intention is not to ignore the findings of the ConLiGen studies (indeed, as noted below, as members of the Consortium, we will be applying to access the full dataset to run the new algorithms). However, we first wanted to work more on the syntax/procedures for the machine learning algorithm. As such, we decided to base this proof-of-concept study on "plausible" candidate genes for response to lithium. As our lab has already published several publications about circadian genes, we know that these genes were relevant to be tested (PMIDs: 33781139, 28095742, 27324142, 25989161, 24716566).

We fully agree that these genes are not the only candidates, and probably not the most obvious ones, given previous results of GWAS in response to lithium. However, the selected candidate genes were available to us in a well-described dataset, and the primary goal of this study was to test the new algorithm, to see whether this approach was feasible. In the first instance, our more modest ambition of this study was to 1) select what we called 'plausible' candidate genes, 2) select a few SNPs within these genes and 3) identify any variations of association results when the definitions of the response to Li phenotypes vary. So, this study should be considered as a first step before moving to the application of the 'new' definition of response to Li to larger datasets and other candidate genes.

As we are members of the ConLiGen consortium, we will be looking to submit an application for such a study (as highlighted by the reviewer) and, assuming we are able to publish this proof of principle study, we would be optimistic that the ConLiGen research committee would approve access to that database in order 1) to identify genetic signals (highly significant, but also significant at a lower level of selection) and 2) to apply similar methods to explore whether highly significant results were still observed (and maybe with higher significance, eg for ERBB2 region), but also to explore whether more minor signals (those not reaching the GWAS significant threshold) might be maximized by the new definition.

However, our view was that, before moving to this wider approach, a proof-of-concept study was desirable, even if the selection of genes and SNPS was simpler (and maybe simplistic), as we wanted to demonstrate the feasibility of our approach.

Given the importance of the reviewer's comment, we have added the following sentences in the limitation section. Our selection of candidate genes is by definition debatable and other approaches may have focused rather on genes that reached (or nearly reached) the GWAS significant threshold in previous GWAS studies (PMC5179929).

 Comment 4

Methods; Lines 337 – 338: Please provide more detail on how the PPV and NPV values were generated.

 Response 4

We described this approach in more detail in an earlier draft of the paper, but as it is of limited utility in interpreting the study findings, we had abbreviated this section (to reduce the word count). However, we are happy to re-introduce the extended text, which states:

We compared estimates of Li response using the original and new approaches to rating the Alda scale, reporting the positive and negative predictive values (PPV, NPV), the overall accuracy and discordance rates. For the purposes of the analyses, we assumed the original ratings represent the ‘gold standard’ (i.e., for categorical outcomes, false positives are cases that were classified as GR according to the new algorithms but not the original rating). The classification obtained for Alda Categories was compared with Algo, whilst the A score/Low B measure was compared with GR according to the Algo (with analyses undertaken using the programme that is publicly available on the Oxford University evidence-based medicine website: https://www.cebm.ox.ac.uk). To interpret the findings, we used the indicators established for diagnostic test comparisons used in clinical settings, which suggest we would expect the new Alda ratings to show PPV, NPV, and accuracy estimates of >80-85% (compared with established ratings).

Comment 5

Methods; Lines 332 – 335- Why were only three SNPs retained?

Response 5

Before the start of this analysis, we were aware about the issue of sample size versus number of parameters to be tested (several definitions of the response, several genes, and several SNPs within genes). The idea beyond the selection of a limited number of genes and a limited number of SNPs was to deal with the issue of correction for multiple testing. As such, we have restricted to one SNPs per gene in 3 genes for 5 different phenotype definitions (thus leading to a corrected p value 0.003).  We have described in the method section how the selection of one SNP per gene has been done in the following sentence. "In order to select the SNPs, we used PLINK to determine the associations between SNPs and the Alda total score. Briefly, for each gene, the program: (1) determines which SNPs are in linkage disequilibrium (LD) (above a certain threshold, fixed in this study at r2 =0.05); (2) performs a single SNP association analysis using a non-parametric approach; and (3) determines the associated SNP having the highest statistical significance (selection of the 'best SNP' option). This method allowed to identify, among all available SNPs within a given gene, which SNP was the most strongly associated with the phenotype". We are aware that this approach (selection of genes followed by the selection of one SNP per gene) deviates from what is classically done in genetic studies. However, we keep in mind that this study does not aim at providing any definitive answer about the associations between studied SNPs and the response to Li, but rather aims at demonstrating that the approach is feasible and, more important, of potential relevance since the signals obtained varied across response definitions. Since this information is already contained in the method section, we did not modify the current text.

Comment 6

Would be good to include a discussion on sample size and power calculation

Response 6

A design with group sample sizes (based on genotypes distribution) can detect effect sizes of 0.5 for (TIMELESS and PPARGC1A) and an effect size of 0.6 for RORA with a power above 0.80 (0.80-0.87), assuming a two-sided criterion for detection (based on means and SDs of the Alda total score) that allows a maximum Type I error rate of alpha=0.05.

We have added the following sentences in the limitations section.

This sample may be regarded as too small to detect any differences between genotypes-based groups for response to Li. However, a design with group sample sizes (based on genotypes distribution) can detect effect sizes of 0.5 for (TIMELESS and PPARGC1A) and an effect size of 0.6 for RORA, with a power above 0.80, assuming a two-sided criterion for detection (based on means and SDs of the Alda total score) that allows a maximum Type I error rate of alpha=0.05.

 Comment 7

Results & Discussion. Well compiled. I appreciated the results expansion on explanation of CHAID. The discussion was well written and not overarching with respect to conclusions.

Response 7

We thank you for this comment.